# Boosting Immunity Through Nutrition and Gut Health: A Narrative Review on Managing Allergies and Multimorbidity

**DOI:** 10.3390/nu17101685

**Published:** 2025-05-15

**Authors:** Eleni Andreou, Christos Papaneophytou

**Affiliations:** Department of Life Sciences, School of Life and Health Sciences, University of Nicosia, 2417 Nicosia, Cyprus; andreou.el@unic.ac.cy

**Keywords:** gut microbiota, immune resilience, allergic diseases, multimorbidity, personalized nutrition

## Abstract

The increasing global burden of allergic diseases and multimorbidity underscores the urgent need for innovative strategies to strengthen immune health. This review explores the complex relationships among nutrition, gut microbiota, immune regulation, allergic diseases, and multimorbidity. It highlights how targeted nutritional and microbial interventions may influence disease outcomes. Dietary components and microbial metabolites dynamically modulated immune function, highlighting the critical role of the gut–immune–metabolism axis in disease pathogenesis and management. Personalized nutrition, guided by advances in diagnostics such as component-resolved diagnostics, basophil activation tests, and epigenetic biomarkers, allows for precise dietary interventions tailored to individual allergy phenotypes and multimorbidity profiles. The Mediterranean diet, breastfeeding, and microbiota-targeted therapies have emerged as effective strategies to enhance immune resilience, reduce inflammation, and manage allergic reactions. Technological advancements, including artificial intelligence-driven dietary assessments, wearable devices, and mobile applications, have further revolutionized personalized dietary management, enabling real-time, precise nutritional monitoring and intervention. Despite these advances, challenges in implementing personalized nutrition persist, including variability in dietary patterns, cultural and socioeconomic factors, and accessibility concerns. Future research should focus on long-term interventional and longitudinal studies to validate precision nutrition strategies and enhance clinical applicability. This integrative approach, combining nutrition, microbiome science, technology, and personalized healthcare, holds substantial promises for sustainable disease prevention and enhanced immune resilience across diverse populations.

## 1. Introduction

The immune system comprises a complex network of cells, signaling molecules, and tissues. Together, they protect the body from various threats, including pathogens, toxins, and abnormal cells like cancer cells [1]. It functions across multiple anatomical sites—including the skin, respiratory tract, and gastrointestinal tract—which serve as the body’s first line of defense through physical and chemical barriers [2]. A well-functioning immune system is essential for survival. This requires the ability to continuously distinguish self from non-self components and to differentiate harmful agents, such as pathogens, from innocuous foreign substances, like dietary antigens [3]. This precision is vital in preventing inappropriate immune responses, including autoimmune diseases and allergies.

Immune responses are broadly categorized into innate and adaptive immunity based on their speed and specificity [4]. Innate immunity offers a rapid, non-specific response to foreign invaders and functions independently of previous exposure, lacking immunological memory [5]. In contrast, adaptive immunity is antigen-specific and involves a delayed activation phase, yet it confers immunologic memory that allows for faster and more targeted responses upon subsequent encounters [6]. While these immune mechanisms are crucial for host defense, their dysregulation contributes to a spectrum of disorders, including allergic and chronic inflammatory diseases [7].

In recent decades, the global prevalence of allergic diseases has risen sharply, posing a growing public health challenge. Approximately 1 billion people currently suffer from allergies, with projections estimating this number could rise to 4 billion within the next 30–40 years [8]. Allergies are hypersensitivity reactions triggered by immune responses to specific antigens known as allergens [9]. Common allergens—including pollen, dust mites, animal dander, and certain foods—often provoke immunoglobulin E (IgE)-mediated responses, leading to T-helper 2 (Th2) polarization, allergen-specific IgE production, and sensitization of effector cells [10]. Repeated allergen exposure activates mast cells and basophils, leading to the release of inflammatory mediators. These can trigger symptoms ranging from mild sneezing and skin irritation to severe respiratory distress and anaphylaxis [11]. The World Health Organization (WHO) recognizes numerous allergic conditions [12]. These include asthma, rhinitis, conjunctivitis, anaphylaxis, atopic eczema, urticaria, and angioedema, as well as hypersensitivity to foods, medications, and insect stings. Notably, gut microbiota has emerged as a key factor influencing allergic disease development, although the precise mechanisms remain under investigation [13].

Alongside the global rise in allergic diseases, multimorbidity—defined as the coexistence of two or more chronic conditions—has emerged as a major public health concern, particularly among aging populations. The presence of multiple chronic illnesses complicates clinical management and substantially elevates the risks of disability, hospitalization, and mortality [14,15].

Multimorbidity typically begins with the acquisition of a single chronic illness, which over time may be followed by the onset of additional conditions. Many chronic diseases, including obesity, type 2 diabetes, cardiovascular disorders (CVDs), autoimmune diseases, and neurodegenerative conditions, are underpinned by a common denominator: chronic low-grade inflammation [16,17]. This persistent inflammatory state is often driven by immune dysregulation, alterations in the gut microbiota, and metabolic imbalances, highlighting the tight interconnection between immune health and multimorbidity [18].

The interplay between multimorbidity and immune dysfunction represents a critical area of research [19]. As immune homeostasis declines, chronic inflammation, cytokine imbalances, and oxidative stress create a biological environment conducive to the development and progression of multiple chronic diseases [19]. This relationship is bidirectional. Immune dysregulation contributes to chronic illness, which in turn, exacerbates immune dysfunction—forming a feedback loop that complicates prevention and treatment. Addressing these interactions may open new therapeutic avenues targeting shared immunological mechanisms. Within this context, nutrition has gained attention as a key modifiable factor capable of influencing immune regulation [20].

Emerging evidence points to gut microbiome composition and dietary patterns as major determinants of immune balance and disease progression in individuals with multimorbidity [21,22,23]. The growing field of immunometabolism—exploring the intersection of immune function and metabolic pathways—has shed light on how nutrients, microbial metabolites, and dietary patterns modulate inflammation and immune aging [24]. Nutritional strategies that include anti-inflammatory diets, probiotic interventions, and adequate intake of essential micronutrients have shown promise in regulating immune responses and potentially reducing the burden of multimorbidity [25].

The strong connection among nutrition, gut health, and immunity has sparked growing interest in dietary and microbiome-targeted interventions. These strategies may enhance immune resilience, prevent allergic sensitization, and reduce chronic disease risk. Traditionally, allergies and chronic diseases have been treated as separate entities, but they share common immunological roots. Allergic disorders often result from inappropriate immune responses to harmless environmental antigens. These conditions frequently co-occur in a sequence known as the “atopic march”, which includes eczema, food allergies, allergic rhinitis, and asthma [26]. This clustering exemplifies multimorbidity and may even extend beyond classical allergic phenotypes.

Support for this integrated view comes from research demonstrating shared immune dysregulation across both allergic and chronic disease states. Allergies are often mediated by type I hypersensitivity, involving IgE and mast cell activation [27]. In contrast, chronic diseases typically feature persistent low-grade inflammation, autoimmunity, or impaired tissue repair [28]. What unites these conditions is a breakdown in immune homeostasis—a central theme explored throughout this review. The rising global burden of allergic diseases and multimorbidity has significant implications for healthcare systems and public health policy. These conditions often coexist, diminishing the quality of life across age groups. Though they may appear distinct, allergies and chronic diseases such as cardiovascular disease, obesity, and neurodegeneration frequently share underlying immunological and microbial disruptions. These include inflammaging, immune senescence, and gut dysbiosis, i.e., diminished microbial diversity and altered microbial metabolism [29,30].

As diet and gut health emerge as key regulators of immune function, they offer novel targets for intervention. Nutrients influence immune activity not only by serving as substrates, but also by modulating epigenetic processes, cytokine responses, and microbial balance. However, most existing studies continue to treat allergic diseases and chronic conditions in isolation. A more unified perspective, situating these within the gut–immune–nutrition axis, is urgently required.

This narrative review examines the mechanistic links among nutrition, the gut microbiota, and immune function, with a particular focus on their roles in allergy and multimorbidity. Drawing from current evidence in immunology, nutrition science, microbiome research, and chronic disease epidemiology, it aims to clarify how dietary and microbial interventions can modulate immune regulation and disease progression. By framing allergies and multimorbidity within a shared biological and immunological context, the review highlights opportunities for integrated prevention and personalized treatment strategies.

## 2. Methods

### 2.1. Search Strategy

This review followed the guidelines of the Scale for the Assessment of Narrative Review Articles (SANRA) [31] to enhance methodological rigor and transparency. A targeted literature search was conducted using the databases PubMed, Scopus, and Web of Science, covering publications up to February 2025. The search used Boolean operators to combine terms such as “nutrition”, “gut microbiota”, “immune system”, “multimorbidity”, “allergies”, and “personalized nutrition”. Additional search terms included “dietary assessment”, “precision medicine”, “microbiome”, “Mediterranean diet”, and “immunonutrition” to broaden the scope and ensure thematic completeness.

### 2.2. Eligibility Criteria

We included peer-reviewed articles published in English that explored the intersections of nutrition, gut health, immune modulation, and chronic disease outcomes. Eligible studies comprised original research (both clinical and mechanistic), systematic reviews, meta-analyses, expert consensus documents, and selected high-impact narrative reviews. The exclusion criteria were as follows: (i) articles not published in English, (ii) preclinical animal-only studies lacking translational relevance, and (iii) abstracts, conference proceedings, and unpublished dissertations.

### 2.3. Study Selection Process

Titles and abstracts were initially screened for relevance, followed by full-text examination. Both authors independently reviewed candidate articles. Discrepancies were resolved through discussion and consensus. Reference lists of key articles were manually scanned to identify any additional pertinent studies.

### 2.4. Quality Appraisal

While this is a narrative review, we conducted a brief critical appraisal of included sources. For narrative reviews and opinion pieces, the SANRA checklist [31] was used to assess strengths and weaknesses (e.g., coverage of literature, critical perspective). For empirical studies, we considered three guiding questions: (i) Was the study design appropriate for the research question? (ii) Are the data and methodology clearly described? (iii) Do the conclusions follow logically from the data presented? Studies with significant methodological flaws were excluded unless cited for historical context or conceptual framing.

### 2.5. Data Synthesis

Key study details (e.g., study population, dietary exposure, immune or clinical outcome) were extracted and organized thematically. Thematic categories were identified inductively based on common patterns across studies, focusing on (i) the gut–immune–nutrition axis, (ii) multimorbidity and inflammatory regulation, and (iii) emerging interventions such as microbiome-targeted and precision nutrition strategies. This approach facilitated the synthesis of complex, heterogeneous evidence into a structured narrative.

## 3. Fundamentals of Immune Function and Regulation

### 3.1. Innate and Adaptive Immunity.: Two Sides of the Same Coin

As mentioned above, the immune system operates through two primary mechanisms: innate immunity, which delivers an immediate, non-specific defense, and adaptive immunity, which provides a highly targeted response but requires developmental time (Table 1).

While traditionally viewed as separate arms of immunity, these systems are now recognized as deeply interconnected, with numerous shared components facilitating coordinated immune activity.

Innate immunity represents the body’s first line of defense, responding rapidly to pathogens without the need for prior sensitization. It is antigen-independent and lacks immunological memory, meaning it responds in the same way with each exposure. This defense relies on a multifaceted network of protective mechanisms: physical and chemical barriers (such as skin, mucosal surfaces, gastric acid, and antimicrobial peptides), cellular components (including neutrophils, monocytes, macrophages, dendritic cells, and natural killer (NK) cells), and soluble factors (such as complement proteins, cytokines, and acute-phase proteins) [34]. While its speed and breadth are beneficial, the non-specific nature of innate immunity can lead to collateral tissue damage [35]. Nonetheless, its evolutionary conservation across species underscores its fundamental role in host survival [36].

Adaptive immunity, by contrast, is characterized by antigen specificity and immunologic memory [37]. It is initiated by antigen-presenting cells (APCs)—notably dendritic cells and macrophages—that process and present pathogen-derived antigens to lymphocytes. Adaptive immunity consists of two major components: B lymphocytes, which mediate humoral immunity through the production of antigen-specific antibodies, and T lymphocytes, which orchestrate cellular immunity [33]. Among T cells, CD4^+^ helper T cells coordinate immune responses via cytokine signaling, while CD8^+^ cytotoxic T cells directly eliminate infected or abnormal cells. Following initial exposure, memory B and T cells persist in circulation, enabling a more rapid and robust response upon re-exposure to the same antigen [38]. Though powerful and precise, adaptive immunity requires tight regulation to avoid pathological consequences, such as autoimmunity or chronic inflammation.

Despite their distinct functional roles, innate and adaptive immunity are highly integrated. Dendritic cells are key to this interface, acting as APCs that initiate T cell responses based on signals received from the innate immune environment [39]. Natural killer T (NKT) cells, which exhibit traits of both NK cells and T cells, serve as a bridge by producing cytokines that modulate both innate and adaptive pathways [40]. Similarly, the complement system, traditionally associated with innate immunity, also plays a pivotal role in enhancing adaptive responses by promoting antigen presentation and B cell activation [41]. Together, these elements exemplify the dynamic crosstalk and coordination between innate and adaptive immunity that is essential for maintaining immune balance and host defense.

### 3.2. Allergies: A Battle of Balance in the Immune System

The immune system plays a pivotal role in differentiating between harmful and harmless substances. In allergic diseases, however, this critical balance is disrupted, resulting in exaggerated immune responses to typically innocuous environmental antigens known as allergens [42]. Historically, allergies were broadly defined as an altered capacity of the body to react to foreign substances; today, the concept has evolved into a more precise definition—a disease characterized by inappropriate immune responses toward normally benign antigens [43].

At the heart of allergic disorders lies immune dysregulation involving both innate and adaptive immunity. Allergic responses are predominantly driven by an exaggerated T-helper 2 (Th2) immune profile, characterized by excessive production of IgE antibodies [44]. Upon re-exposure to the allergen, these allergen-specific IgE antibodies bind and activate mast cells, triggering the release of inflammatory mediators such as histamine, leukotrienes, and cytokines [45]. These mediators produce classic allergy symptoms, including sneezing, itching, airway constriction, and inflammation, ranging from mild discomfort to severe asthma exacerbations or potentially fatal anaphylaxis [46]. Under physiological conditions, regulatory T cells (Tregs) maintain immune tolerance and prevent unnecessary immune activation [47]. In individuals with allergic diseases, however, Treg function is often compromised, with decreased production of anti-inflammatory cytokines like interleukin-10 (IL-10) and transforming growth factor beta (TGF-β) [48]. This impaired regulation contributes to the dominance of Th2-driven responses, exacerbating allergic inflammation and hypersensitivity. Persistent activation of key immune effectors—including mast cells, eosinophils, and basophils—and their subsequent release of inflammatory mediators sustain and intensify allergic pathology (Table 2).

Globally, allergic diseases represent a growing public health concern, with prevalence increasing significantly over recent decades [50]. This rise is driven by a complex interplay of genetic and environmental factors, including immune dysregulation, alterations in gut microbiota, infectious exposures, and lifestyle changes associated with modernization. Atopy—a genetic predisposition characterized by heightened IgE responses to minor environmental stimuli—is a well-established risk factor contributing to allergic diseases [51]. Common allergens include aeroallergens (e.g., pollen, dust mites, mold spores), food allergens (e.g., nuts, eggs, shellfish), and chemical allergens (e.g., dyes, fragrances, preservatives).

Understanding immune dysfunction in allergies is crucial for developing effective therapeutic strategies. Emerging treatments such as allergen-specific immunotherapy, biologics targeting key Th2 cytokines, and microbiota-based interventions represent promising approaches for restoring immune balance and alleviating allergic disease symptoms.

### 3.3. The Overlapping Puzzle: Understanding Multimorbidity and Immune Dysregulation

Multimorbidity, defined as the coexistence of two or more chronic diseases within an individual, has emerged as an increasingly prevalent clinical reality, especially among aging populations [52]. In some cases, multimorbidity is recognized as a distinct clinical entity—for example, metabolic syndrome (MS) [53]. MS is diagnosed when central obesity coexists with at least two additional factors: elevated triglycerides, reduced high-density lipoprotein (HDL) cholesterol, elevated blood pressure, or elevated fasting plasma glucose [54]. This syndrome is strongly associated with CVDs, and both MS and CVDs significantly contribute to biological aging through mechanisms involving oxidative stress and chronic low-grade inflammation, known as inflammaging [55,56].

Chronic inflammation, a hallmark of aging, is closely associated with multimorbidity. Elevated inflammatory markers such as interleukin-6 (IL-6), tumor necrosis factor-alpha (TNF-α), and C-reactive protein (CRP) are commonly observed in individuals living with multiple chronic conditions [57]. Notably, higher inflammatory marker levels are linked to an increased risk of disability, hospitalization, and mortality [58]. Consequently, inflammatory biomarkers have become integral to aging biomarker panels in clinical trials [59].

In a recent study, Sauver et al. [60] demonstrated significantly elevated levels of IL-6 and tumor necrosis factor-alpha (TNF-α) in individuals with higher multimorbidity percentiles, particularly among women and adults aged 70 years and older. Interestingly, IL-10, an anti-inflammatory cytokine, showed no correlation with multimorbidity, suggesting that pro-inflammatory pathways, rather than compensatory anti-inflammatory responses, predominantly drive multimorbid conditions.

Recent advances have expanded our understanding of multimorbidity beyond traditional inflammatory markers. Chen et al. [61] conducted a large-scale proteomics analysis involving over 53,000 adults, identifying 972 proteins shared across multiple chronic diseases and 345 proteins uniquely linked to specific conditions. Their findings emphasize inflammation’s central role in multimorbidity, with transcription factors such as nuclear factor NF-kappa-B (NFKB1), JUN, and RELA identified as upstream regulators. Additionally, proteins including GDF15, PLAUR, WFDC2, and AREG were associated with the risk of multiple chronic conditions, highlighting their potential as biomarkers for multimorbidity progression. These results underscore the necessity of developing targeted interventions aimed at inflammatory and molecular mechanisms underlying chronic disease clustering.

Multimorbidity represents more than the simple accumulation of separate diseases; it involves a complex interplay of shared biological and immunological pathways [62]. The concept of inflammaging offers a valuable framework to understand how chronic low-grade inflammation accelerates disease progression, contributing to functional decline and reduced life expectancy [30,63]. Studies such as the Mid-Life in the United States (MIDUS) [64] and the InCHIANTI study [65] have demonstrated that circulating IL-6 and CRP levels rise proportionally with increasing numbers of chronic conditions, and that individuals with higher baseline IL-6 levels experience a steeper trajectory of disease accumulation over time.

Despite the growing recognition of multimorbidity as a pressing health challenge, effective preventive and therapeutic strategies remain limited. Current approaches typically emphasize lifestyle modifications, inflammation-targeting treatments, and precision medicine. However, as emerging evidence suggests, future therapeutic strategies must directly target the shared molecular and immunological dysregulation underlying multimorbidity to improve health outcomes.

#### Multimorbidity and Immune Aging: The Chronic Loop

While inflammaging is widely recognized as a hallmark of aging and a driver of multimorbidity, emerging research underscores that immune aging encompasses more than just cytokine dysregulation. Instead, immune aging represents a multidimensional decline characterized by several interconnected processes, as discussed below.

Immunosenescence—loss of immune surveillance: Aging is accompanied by an accumulation of senescent immune cells, particularly CD8^+^ T cells and memory T cells. These cells exhibit reduced proliferative capacity and actively secrete pro-inflammatory mediators known collectively as the senescence-associated secretory phenotype (SASP). The accumulation of these cells not only exacerbates systemic inflammation but also disrupts tissue repair, weakens tolerance mechanisms, and increases susceptibility to autoimmune disorders [66].Neuroimmune crosstalk and cognitive decline: Recent evidence highlights neuroimmune interactions as crucial to aging-related diseases [67]. For example, activated microglia and elevated levels of inflammatory cytokines, such as IL-1β and IL-6, impair synaptic plasticity, contributing to cognitive impairment and conditions like Alzheimer’s disease and depression. Notably, these neurological conditions often co-occur with cardiometabolic disorders, underscoring their interconnected pathogenesis in aging populations [68].Metaflammation and mitochondrial dysfunction: Metaflammation—a chronic, low-grade inflammatory response triggered by metabolic overload and overnutrition—differs fundamentally from classical inflammation. Metabolic stress induces mitochondrial dysfunction, increasing reactive oxygen species (ROS) generation and mitochondrial DNA release, subsequently activating inflammasomes such as NLRP3. This process contributes significantly to metabolic syndrome and insulin resistance, two key components of multimorbidity [69].Epigenetic aging and biological clocks: Biological aging, measured by DNA methylation clocks (e.g., the Horvath clock), correlates more closely with multimorbidity risk than chronological age alone. Epigenetic drift accelerates under conditions of chronic inflammation, lifestyle factors, and microbiota alterations, forming a mechanistic bridge linking immune aging with metabolic and cardiovascular diseases [70].

## 4. Microbial Guardians: How the Gut Shapes Our Immunity

The human microbiota consists of diverse bacteria, fungi, viruses, and archaea that reside on and within our bodies, dynamically interacting with their host and surroundings. The term “microbiome” further includes these microorganisms’ collective genomes and the physicochemical properties of their niche. These microorganisms collectively shape numerous physiological processes, influencing host health in diverse and significant ways. Depending on their effects, microbes can be categorized as symbiotic (beneficial), pathogenic (harmful), or neutral, with neutral microbes capable of transitioning between beneficial and harmful states depending on the host’s condition and environment [71,72]. In 2001, at the behest of the Food and Agriculture Organization of the United Nations and with support from the WHO, an expert panel was assembled to refine the definition of probiotics. The ensuing document characterized probiotics as “*Live microorganisms which when administered in adequate amounts confer a health benefit on the host*” [73]. Later, in 2014, a consensus panel revisited this definition and made a minor modification, substituting “*which*” with “*that*” [74].

Central to these interactions is the gut microbiota, a complex and dynamic microbial ecosystem that plays a pivotal role in maintaining host health. It influences immune function both directly, through interactions with immune cells, and indirectly, by producing metabolites from dietary components [75,76]. In particular, the gut microbiota is essential for shaping mucosal immunity, modulating antigen presentation, and regulating systemic immune balance through its metabolic byproducts [76].

The indispensable roles of commensal microbes at mucosal surfaces in immune regulation, influencing immune system development and homeostasis, are increasingly recognized [77]. Their role in shaping immune responses across various organs, modulating type 2 immunity, regulating basophil hematopoiesis, and maintaining the integrity of epithelial barriers has been highlighted [78].

Bacterial metabolites produced by gut microbiota significantly influence immune maturation and function. Short-chain fatty acids (SCFAs), such as butyrate and propionate, are prominent examples. SCFAs exert immunoregulatory effects, notably by modulating FOXP3^+^ regulatory Tregs, which are crucial for maintaining immune tolerance and preventing inflammation-induced damage. Thus, effective crosstalk between mucosal immune cells and resident microbiota is vital to maintaining balanced immune responses, optimizing protection without triggering excessive inflammatory reactions [79,80].

Disruptions in microbial dysbiosis are increasingly linked to immune-mediated disorders, including allergic diseases [81]. Altered microbiota composition can induce immune dysregulation, increasing susceptibility to inflammatory and hypersensitivity reactions. These insights underscore the importance of preserving a balanced gut microbiome for robust immune resilience [82].

Interestingly, the microbiome also impacts cognitive function, highlighting its systemic reach beyond gastrointestinal immunity [83]. This dual capacity of the microbiome—to promote immune tolerance or provoke inflammation—illustrates its intricate involvement in maintaining immune equilibrium and its potential role in disease pathogenesis.

Dynamic interactions between the gut microbiota and the host’s innate and adaptive immune systems are fundamental for intestinal homeostasis and inflammatory prevention [84]. This complex ecosystem, composed of bacteria, fungi, and viruses, exists in a symbiotic relationship with its host. The gut microbiota participates actively in immune responses by metabolizing dietary proteins and carbohydrates, synthesizing essential vitamins, and producing bioactive compounds that mediate the dialog between gut epithelial cells and immune cells [85]. External factors such as diet, antibiotics, environmental exposures, and lifestyle significantly influence microbiota composition, highlighting its adaptable nature.

Integral to the gut’s defense system is the intestinal epithelial barrier, reinforced by mucus layers, secretory IgA, and antimicrobial peptides, functioning as a selective filter to separate microbiota from host immune cells [86]. Dysregulated interactions between the gut microbiota and mucosal immunity can disrupt this barrier, leading to increased gut permeability (“leaky gut”), dysbiosis with a rise in pathogenic Gram-negative bacteria, metabolic disturbances, and heightened susceptibility to infections and chronic inflammatory conditions [82].

Beyond its local effects, the gut microbiota plays a foundational role in systemic immune system development and maturation. Accumulating evidence demonstrates that commensal bacteria are critical for immune organ development, shaping immune response patterns, influencing oral tolerance mechanisms, regulatory T cell development, and host metabolism regulation [87,88]. Notably, disruptions in gut microbiota composition are strongly associated with allergic diseases, underscoring its significant role in systemic inflammation and immune-mediated disorders [89].

Given microbiota’s central role in immune modulation, dietary interventions, probiotics, and microbiome-based therapeutic strategies have become major research areas aimed at restoring microbial equilibrium, reinforcing immune homeostasis, and preventing immune-related diseases.

### Allergies and Gut Health: The Microbiome–Immune Connection

The rising prevalence of allergic diseases in recent decades has spurred interest in exploring the environmental and microbial factors influencing immune responses. A prominent theory, the hygiene hypothesis, first proposed by Dr. Strachan in the late 1980s [90], posits that reduced microbial exposure in early life, due to heightened sanitation, antibiotic overuse, and Western dietary habits, may alter immune maturation and predispose individuals to allergic inflammation. Epidemiological evidence supports the notion that high microbial diversity in utero and during early life plays a protective role against the development of allergic diseases [91].

Allergic diseases, which encompass respiratory, cutaneous, and food allergies, typically involve a dominant Th2 immune response. This response triggers the production of IL-4, IL-5, and IL-13, which facilitate IgE class switching, eosinophilic inflammation, and mast cell activation, characteristic of allergic reactions [92]. Furthermore, T cell subsets such as Th9 cells exacerbate allergic inflammation by secreting IL-9 and IL-10 [93].

The importance of the microbiome as a central regulator of immune function, crucial for immune development, antigen tolerance, and inflammation modulation, is increasingly recognized [94]. While the airway microbiome directly modulates local inflammatory responses in conditions like asthma, the gut microbiota exerts systemic effects that influence susceptibility to allergic diseases [95]. Changes in diet and environment that lead to gut dysbiosis can significantly affect microbial composition and metabolic activity, affecting not only gut health but also systemic immune responses [96].

The gut microbiota’s role extends beyond mere local effects; it is instrumental in immune system development and maturation across life stages. The microbial diversity of the gut, which is relatively low in infancy, increases through childhood and adulthood due to various dietary, environmental, and immunological factors [97,98]. Among the microbial metabolites, SCFAs, bile acid conjugates, and tryptophan metabolites are particularly significant for their roles in modulating allergic responses [99]. SCFAs, such as butyrate and propionate, are pivotal in regulating immune homeostasis, influencing the activity of colonic FOXP3+ regulatory T cells, which are essential for maintaining immune tolerance and mitigating excessive inflammation [100].

Early-life factors, including maternal microbiome composition, mode of birth delivery, and infant feeding, profoundly influence neonatal microbial colonization. Breastfeeding promotes the growth of beneficial bacteria like *Bifidobacterium*, while formula feeding may foster distinct microbial patterns that impact immune system maturation (discussed further below). Additionally, exposure to diverse environmental conditions, such as living on a farm, has been linked to enhanced gut microbiota diversity and reduced risk of allergic diseases, supporting the hygiene hypothesis [101,102]. For example, the GABRIELA (Multidisciplinary Study to Identify the Genetic and Environmental Causes of Asthma in the European Community [GABRIEL] Advanced Study) and PARSIFAL (Prevention of Allergy-Risk Factors for Sensitization in Children Related to Farming and Anthroposophic Lifestyle) studies found that children raised in farm environments—exposed to diverse microbial communities—exhibited lower rates of asthma and allergies compared to their urban counterparts [103,104].

One compelling link between gut microbiota and allergic inflammation involves the role of dietary fermentable fibers in influencing microbial metabolism. Trompette et al. [105] showed that a high-fiber diet modifies the gut and lung microbiota composition, particularly altering the Firmicutes/Bacteroidetes ratio. The fermentation of dietary fibers by gut microbes enhances the production of SCFAs, which have immunomodulatory effects on host immune cells. Mice on a high-fiber diet displayed elevated circulating SCFAs and were protected against allergic airway inflammation, whereas those on a low-fiber diet had decreased SCFA levels and increased susceptibility to such conditions. Further analysis indicated that propionate, mediated through G-protein-coupled receptor 41 (GPR41, also known as FFAR3), plays a crucial role in modulating immune responses and allergic disease progression.

The intricate role of the gut microbiota in modulating allergic risk is supported by numerous epidemiological and mechanistic studies. Birth cohort studies further reveal that decreased *Bifidobacterium* and *Lactobacillus* abundance correlates with atopic dermatitis risk [106] while reduced butyrate-producing bacteria are linked to asthma development [107], and early colonization by *Clostridioides difficile* is associated with increased allergic sensitization [108].

Host–microbiota communication occurs largely through pattern recognition receptors (PRRs), such as Toll-like receptors (TLRs), which detect microbial-associated molecular patterns. The activation of TLR4 supports dendritic cell maturation and Treg development, whereas disruption of this pathway—as seen in MyD88-deficient mice—exacerbates allergic inflammation [109,110]. Commensals further promote Treg differentiation via SCFA production, *Bacteroides fragilis* polysaccharide A (PSA), and the induction of epithelial-derived TGF-β by *Clostridial* clusters [111].

Beyond Treg induction, microbial signals influence the Th1/Th2 balance. A Th2-skewed response underpins allergic diseases; however, exposure to specific microbes can enhance Th1 and Th17 responses that counteract Th2 dominance. For instance, *Segmented Filamentous Bacteria* (SFB) promote Th17 differentiation [112], while *Bifidobacterium* facilitates Th1 polarization via dendritic cell priming [113]. Notably, *Helicobacter pylori* colonization in early life has been inversely associated with asthma risk, possibly through Treg induction and immune tolerance [114].

The gut microbiota also contributes to epithelial barrier integrity. Commensals like *Akkermansia muciniphila* enhance mucin production and tight junction assembly, while butyrate supports barrier function through tight junction protein regulation [115,116]. In contrast, increased intestinal permeability—a feature of food-allergic individuals—has been associated with dysbiosis and reduced SCFA levels [117].

Therapeutic modulation of the microbiota through probiotics, prebiotics, and synbiotics has shown promise, albeit with variable results. The Probiotics in Pregnancy Study (PiP Study) [118] investigated whether maternal supplementation with *Lactobacillus rhamnosus* HN001 from early pregnancy through breastfeeding could reduce infant eczema and atopic sensitization by one year. The study found that probiotic supplementation was associated with a significant reduction in eczema prevalence, supporting the potential role of early microbial interventions in allergy prevention. Additionally, maternal health benefits included lower rates of gestational diabetes mellitus, bacterial vaginosis, and Group B Streptococcal vaginal colonization before birth, as well as reduced postpartum depression and anxiety. Lyons et al. [119] demonstrated that bacterial strain-specific induction of Foxp3+ T regulatory cells plays a crucial role in protecting against allergic inflammation in murine models. Their study revealed that *Bifidobacterium longum* AH1206 significantly increased Foxp3+ T regulatory cell numbers in infant, adult, and germ-free mice, while altering gene expression in Peyer’s patches to reduce antigen presentation, TLR signaling, and cytokine production, and enhance retinoic acid metabolism. This strain provided protection against airway inflammation and blocked IgE induction in oral allergy models, whereas *Bifidobacterium breve* AH1205 exhibited limited efficacy, inducing regulatory T cells only in infant mice. *Lactobacillus salivarius* AH102 had no impact on T regulatory cell numbers or allergic response. Their findings suggest that specific probiotics may hold promise in mitigating respiratory and dietary allergies by modulating immune regulation. Prebiotics, such as galacto-oligosaccharides (GOSs) and fructo-oligosaccharides (FOSs), reduced allergic outcomes in high-risk infants, and human milk oligosaccharides (HMOs) have been shown to promote beneficial bacterial growth [120].

A randomized double-blind study demonstrated that the synbiotic use of Bifidobacterium breve M-16V in combination with short-chain galacto-oligosaccharides (scGOSs) and long-chain fructo-oligosaccharides (lcFOSs) successfully compensated for the delayed colonization of *Bifidobacterium* in cesarean-delivered infants [121]. This intervention led to a significantly higher proportion of bifidobacteria from the early days of life, accompanied by reduced Enterobacteriaceae levels and a shift towards an acidic gut environment, marked by increased acetate production. These changes emulated the microbiota composition typically observed in vaginally delivered infants, highlighting the potential of synbiotics in supporting early gut health. Furthermore, post hoc analyses suggested a lower incidence of eczema and atopic dermatitis in the synbiotic group, although further research is needed to establish direct clinical benefits [121].

Given the profound impact of gut microbiota in immune regulation and allergic disease development, microbiome-targeted interventions offer promising strategies for allergy prevention and treatment [122]. These interventions include supplementing probiotics and prebiotics to restore microbial balance, modifying diets to enhance fiber intake and polyphenol consumption, and exploring microbiome-based therapies such as fecal microbiota transplantation (FMT) and next-generation postbiotics. As research advances, understanding the intricate relationship among gut microbiota, diet, and immune function will be crucial for developing precision medicine approaches to effectively address the growing burden of allergic diseases.

## 5. Nutrition, Immunity, and Microbiota: The Triad of Resilience

Immune function is not solely dictated by genetic predisposition but is dynamically influenced by dietary patterns and gut microbiota (Figure 1). The gut–immune–metabolic axis is increasingly acknowledged as a pivotal regulator of health, where nutrients and microbial metabolites serve as powerful immunomodulators. This section will delve into specific nutrients, dietary strategies, and metabolic pathways that can be optimized through dietary interventions to bolster immune function and promote long-term health.

### 5.1. Nutrition and Gut Health Hacks for Managing Allergies and Multimorbidity

Dietary components play a crucial role in shaping immune responses, with the immune system intricately connected to nutrition and metabolism [123]. The major dietary components influencing the immune system are summarized in Table 3. Nutrients serve as building blocks for immune cells, regulate cytokine production, and modulate inflammatory pathways through direct metabolic signaling or interactions with the gut microbiota [124]. The complex interplay among nutritional status, microbial composition, and immune function is often referred to as the gut–immune–metabolism axis, highlighting how diet can either enhance immune resilience against pathogens or promote low-grade chronic inflammation and immune dysregulation [125].

Nutrients, both macro and micro, influence immune activity by modulating energy availability, cellular metabolism, and inflammatory responses. Essential micronutrients, such as vitamins A, C, D, E, B6, B12, and folate (discussed further below), alongside vital minerals like zinc, selenium, iron, magnesium, and copper, function as cofactors in critical enzymatic reactions that underpin immune defense and signaling [20]. Additionally, diet-derived microbial metabolites, including SCFAs, polyphenols, and bile acids, act as significant immunomodulators, affecting T cell differentiation, cytokine secretion, and gut barrier integrity [126].

The impact of diet on immune function is significant, with evidence linking anti-inflammatory dietary patterns to enhanced immune capabilities and a reduced risk of chronic diseases [127]. Conversely, diets rich in processed foods, refined sugars, and unhealthy fats are linked to metabolic dysfunction and systemic inflammation. Emerging evidence also supports that intermittent fasting and caloric restriction can further promote immune renewal and metabolic flexibility, potentially mitigating age-related immune decline [128].

**Table 3 nutrients-17-01685-t003:** Key Nutrients for immune function, their benefits, and dietary sources ^1^.

Nutrient	Immune Benefits	Dietary Sources
Vitamin D	Regulates T cell responses, enhances antimicrobial peptides, and reduces autoimmune activity.	Sunlight exposure, fatty fish, fortified dairy, eggs
Vitamin A	Supports mucosal immunity, influences Treg cells, enhances gut barrier defense.	Liver, carrots, sweet potatoes, dark leafy greens
Vitamin C	Antioxidant, crucial for neutrophil function, cytokine production, and oxidative stress reduction.	Citrus fruits, bell peppers, strawberries, broccoli
Vitamin E	Reduces oxidative stress, supports T cell function, enhances NK cell activity.	Nuts, seeds, spinach, sunflower oil
B Vitamins (B6, B12, Folate)	Essential for immune cell proliferation, DNA synthesis, and homocysteine regulation.	Whole grains, legumes, eggs, leafy greens, animal proteins
Zinc	Critical for T cell activation, antioxidant function, and mucosal immunity.	Shellfish, red meat, pumpkin seeds, legumes
Selenium	Supports glutathione peroxidase, reduces inflammation, and oxidative stress.	Brazil nuts, seafood, whole grains, eggs
Iron	Required for immune cell function, but excess can promote oxidative stress and microbial growth.	Red meat, lentils, spinach, fortified cereals
Magnesium	Regulates inflammation, stress response, and mitochondrial function.	Nuts, seeds, whole grains, leafy greens
Omega-3 Fatty Acids (EPA and DHA)	Modulates inflammatory responses, supports gut microbiota diversity, suppresses Th2-driven allergic inflammation.	Fatty fish (salmon, sardines), flaxseeds, chia seeds, walnuts
Polyphenols	Modulates gut microbiota, suppresses oxidative stress, enhances anti-inflammatory pathways.	Berries, tea, dark chocolate, turmeric, grapes
Short-Chain Fatty Acids (SCFAs)	Regulate Treg cells, reduce inflammation, and support intestinal homeostasis.	Fermented fiber-rich foods (oats, legumes, green bananas, resistant starch sources)

^1^ Data obtained from [20,129,130].

A deeper understanding of the interplay among nutrition, metabolism, and immune resilience forms the foundation for precision nutrition and microbiome-targeted interventions aimed at preventing and managing immune-mediated and chronic inflammatory diseases. The role of the immune system, gut microbiota, and microbial exposures in the development and progression of allergic diseases is increasingly recognized. While avoiding allergen exposure is a primary strategy, pharmacological treatments such as steroids, antihistamines, and other symptom-relieving drugs are commonly employed [131]. Nonetheless, dietary and microbiome-focused strategies are emerging as promising adjunctive approaches in allergy management and immune modulation.

### 5.2. Optimizing Immune Function Through Nutrient-Rich Diets

Nutrition is fundamental in regulating immune responses, as it ensures that immune cells are adequately nourished to respond effectively against pathogens and regulate inflammation. A diverse array of micronutrients, minerals, vitamins, and specific macronutrients—including particular amino acids, cholesterol, and fatty acids—play crucial roles in modulating immune activity [20]. These nutrients are essential for initiating rapid immune defenses and preventing excessive chronic inflammation that can lead to immune dysregulation. Inadequate dietary intake or poor nutrient absorption can significantly impair immune system function, thereby increasing susceptibility to infections and chronic inflammatory diseases.

The development and function of the immune system are also influenced by bacterial stimuli, with the gut microbiome playing an integral role in immune maturation [132]. Maintaining a diet rich in essential nutrients is key to supporting the immune system. Consuming a varied diet rich in colorful fruits and vegetables provides vital antioxidants such as vitamin C and beta-carotene, which protect immune cells from oxidative stress. Cruciferous vegetables, such as broccoli, kale, and cabbage, are high in sulforaphane, a compound that activates antioxidant pathways via Nrf2 signaling. Citrus fruits, berries, and bell peppers, known for their high vitamin C content, enhance neutrophil function and strengthen epithelial barrier integrity.

Zinc, found in foods like oysters, pumpkin seeds, and legumes, is crucial for thymulin production, which is essential for T cell maturation [133]. Selenium, abundant in Brazil nuts and seafood, bolsters neutrophil and NK cell activities, contributing to the body’s antioxidant defenses [134]. Vitamin D, prevalent in fatty fish and fortified foods, plays a pivotal role in regulating both innate and adaptive immune responses, crucial for controlling inflammation and modulating overall immune function [135].

#### 5.2.1. The Role of Vitamins in Immune Function

Vitamins are crucial for the development, function, and balance of the immune system, each playing specific roles in bolstering immune responses. Table 4 summarizes the primary vitamins involved in immune function, their specific roles, and practical information regarding their requirements and sources.

Interactions and synergy among vitamins play a crucial role in optimizing immune function, as many micronutrients do not act in isolation but instead work together to support various aspects of the immune response [136]. For example, vitamins C and E demonstrate a classic synergistic relationship—vitamin C helps regenerate oxidized vitamin E, allowing it to continue functioning as a potent lipid-soluble antioxidant. This partnership enhances the body’s capacity to neutralize oxidative radicals, thereby protecting immune cells from oxidative stress, which is particularly heightened during infections and inflammatory responses [137].

Beyond this pairing, several other vitamins support each other’s functions in significant ways. Vitamin D enhances calcium absorption, indirectly supporting immune-related signaling pathways [138]. Folate and vitamin B12 are jointly required for methylation reactions and nucleic acid synthesis, both of which are essential for the replication and repair of rapidly dividing immune cells [139]. Additionally, the interaction of zinc and selenium with antioxidant vitamins supports enzymatic systems that mitigate oxidative stress and inflammation, further enhancing immune functionality.

It has been suggested that a balanced intake across the full spectrum of vitamins, achieved through a nutrient-dense diet, is much more effective than high-dose supplementation of a single vitamin, which can lead to nutritional imbalances or toxicity [140]. The immune system’s resilience depends on the cumulative, complementary actions of multiple nutrients that support innate and adaptive immune mechanisms. This holistic approach to micronutrient support is fundamental during periods of increased immune challenge, such as during infections, aging, or chronic disease states, as highlighted in recent research on COVID-19 and immune preparedness. The synergy of multiple micronutrients, rather than isolated supplementation, offers a more sustainable and physiologically aligned strategy for optimizing immune function [141].

**Table 4 nutrients-17-01685-t004:** Summary of key vitamins, their immune functions, recommended intakes, and food sources.

Vitamin	Primary Immune Functions	Deficiency Effects	Recommended Intake	Dietary Sources	Refs
A	Maintains mucosal immunity Preserves epithelial barrier integrity Supports T cell differentiation Enhances IgA production Supports regulatory T cell function	Increased risk of respiratory infections Weakened mucosal defenses Immune dysregulation	900 μg RAE ^1^/day (men) 700 μg RAE/day (women)	Preformed: Liver, eggs, dairy Provitamin A carotenoids: Carrots, sweet potatoes, dark leafy greens	[142,143]
C	Potent antioxidant Cofactor for biosynthetic and gene-regulatory enzymes Strengthens epithelial barriers Enhances neutrophil chemotaxis and phagocytosis Clears spent neutrophils to limit inflammation Supports lymphocyte differentiation and proliferation	Compromised immune defense Increased susceptibility to infections Levels depleted during infection/inflammation	100–200 mg/day for prevention Higher doses during illness	Citrus fruits Berries Peppers Broccoli Kiwi	[144]
D	Enhances antimicrobial peptide production Promotes maturation of macrophages and dendritic cells Modulates T-helper cell responses Supports regulatory T cell functions	Increased susceptibility to infections Higher risk of autoimmune conditions Respiratory tract diseases	600 IU ^2^/day (≤70 years) 800 IU/day (>71 years) Optimal blood levels: 30–50 ng/mL	Fatty fish Fortified dairy products Egg yolks	[145,146,147]
E	Protects immune cell membranes from oxidative damage Stabilizes immune Enhances T cell proliferation Boosts interleukin-2 production Increases natural killer cell activity	Impaired cell-mediated and humoral immune responses Reduced ability to fight infections	15 mg alpha-tocopherol/day	Nuts (almonds, hazelnuts) Seeds Spinach Sunflower oil Whole grains	[148,149]
B6	Essential for lymphocyte proliferation and differentiation Supports cytokine and antibody production Coenzyme in amino acid metabolism	Impaired cell-mediated immunity Reduced interleukin-2 production	1.3–1.7 mg/day	Poultry Fish Bananas Potatoes	[150,151]
B12	Supports DNA synthesis and immune cell replication Regulates homocysteine levels	Diminished proliferative capacity of immune cells Compromised immune response	2.4 μg/day	Meat Eggs Dairy products Fortified foods	[150,151]
Folate (B9)	Essential for DNA synthesis and repair Supports cell division of T and B lymphocytes Regulates homocysteine metabolism	Reduced immune cell proliferation Increased susceptibility to infections	400 μg DFE ^3^/day	Leafy greens Legumes Citrus fruits Fortified grains	[150,151]

^1^ RAE = Retinol Activity Equivalents; ^2^ IU = International Units; ^3^ DFE = Dietary Folate Equivalents.

#### 5.2.2. The Mediterranean Diet and Allergies: A Protective Role?

The Mediterranean diet (MedDiet or MD) is a dietary model inspired by the traditional eating patterns of certain countries surrounding the Mediterranean basin. While at least 16 nations border the Mediterranean Sea, dietary habits vary not only between countries but also across different regions within the same country. Cultural traditions, ethnic backgrounds, religious practices, economic conditions, and agricultural production influence these variations.

Despite these differences, the Mediterranean dietary pattern (MDP) shares several defining characteristics (Table 5). It is distinguished by the abundant use of olive oil and a high intake of fruits, vegetables, whole grains, legumes, nuts, and seeds. It also includes moderate consumption of fish and shellfish, white meat, eggs, and fermented dairy products such as cheese and yogurt, while red meat, processed meats, and foods high in sugar are consumed sparingly. Additionally, the MDP traditionally encourages regular but moderate consumption of wine, particularly red wine, alongside meals [152].

Several studies have highlighted the potential of the MD in reducing allergic disease risk, although findings remain inconsistent across populations and study designs. A systematic review by Panagiotou et al. [154] assessed the impact of MD components on food allergies and found that maternal adherence to the MD during pregnancy and lactation was associated with a lower risk of food allergies in infants. However, when dietary interventions were introduced solely during pregnancy or restricted to the infant’s early months, the protective effects were less evident. These findings suggest that early-life exposure to bioactive compounds in the MD, particularly through maternal nutrition, may help shape immune tolerance and reduce allergic sensitization. Among the key contributors to this protective effect are polyphenols, omega-3 fatty acids, fiber, and vitamins, which help regulate inflammatory pathways, gut microbiota composition, and immune system function [155].

Another systematic review by Koumpagioti et al. [156] analyzed data on childhood allergy risk and found that while MD adherence was associated with a lower prevalence of asthma, its effects on allergic rhinitis, eczema, and atopy were less conclusive. Similarly, a study conducted on pediatric populations linked higher MD adherence with reduced asthma incidence but found no significant reduction in allergic sensitization [157]. The mixed results across studies highlight the need for randomized controlled trials (RCTs) with standardized dietary assessments and controlled variables to better understand the MD’s role in allergy prevention.

Nevertheless, the MD’s potential in allergy prevention is largely attributed to its rich composition of bioactive nutrients, which collectively modulate immune responses and gut microbiota balance. Several mechanisms have been proposed to explain its protective role:Anti-inflammatory properties: The MD is abundant in anti-inflammatory compounds, including polyphenols, flavonoids, and omega-3 fatty acids, which help regulate immune responses and suppress chronic inflammation. Since allergic diseases are characterized by excessive Th2-driven immune activation and inflammation, the MD’s ability to modulate cytokine production and inhibit oxidative stress may contribute to reduced allergic symptoms [155].Gut microbiota modulation: A well-balanced gut microbiota is essential for immune homeostasis and allergic tolerance [158]. The MD, which is rich in fiber, fermented foods, and plant-based prebiotics, promotes gut microbial diversity and the production of SCFAs. These microbial metabolites enhance Treg cell activity, reduce intestinal permeability, and mitigate systemic inflammation, thereby lowering allergy susceptibility.Antioxidant defense against allergic reactions: Many MD components, including fruits, vegetables, olive oil, and nuts, are rich in antioxidants such as vitamins C and E, carotenoids, and polyphenols. These compounds help protect immune cells from oxidative stress-induced damage, which is a key factor in allergic inflammation and airway hyperreactivity. By neutralizing reactive oxygen species (ROS), these antioxidants may reduce mast cell degranulation and histamine release, thereby lessening the severity of allergic reactions [159].Polyunsaturated fatty acids (PUFAs) and immune modulation: The MD is rich in long-chain omega-3 fatty acids (EPA and DHA) from fish, olive oil, and nuts, which exert immune-modulating effects. PUFAs influence eicosanoid synthesis, leading to the production of anti-inflammatory mediators that help balance Th1/Th2 immune responses [160]. Several studies have shown that higher omega-3 intake is associated with lower asthma prevalence and improved lung function, supporting the hypothesis that the MD may be particularly beneficial in respiratory allergies [161].Maternal nutrition and early immune programming: Maternal diet plays a critical role in fetal immune system development [162]. Studies suggest that adherence to the MD during pregnancy may reduce the risk of allergic sensitization in offspring, possibly through epigenetic modifications, altered gut microbiota transmission, and early exposure to immune-regulating nutrients [163]. However, findings remain inconsistent, emphasizing the need for longitudinal studies tracking maternal and child dietary patterns.

Despite promising findings, the relationship between the Mediterranean diet and allergies remains incompletely understood due to heterogeneity in study designs, dietary assessments, and population-specific factors [164]. Several key challenges contribute to this complexity. Variability in dietary patterns presents a significant obstacle, as the definition and adherence to the Mediterranean diet differ across studies, making direct comparisons difficult. Additionally, allergy risk is influenced by a combination of genetic and environmental factors, including genetic predisposition, air pollution, microbiome diversity, and early-life exposures, all of which interact in intricate ways to shape an individual’s susceptibility to allergic conditions [165]. Long-term adherence and dietary compliance further complicate research efforts, particularly in children and adolescents, where maintaining consistent adherence to the Mediterranean diet remains a challenge in dietary intervention studies. To establish a clearer understanding of the Mediterranean diet’s role in allergy prevention, future research should prioritize large-scale, randomized controlled trials with well-defined dietary interventions. Moreover, personalized dietary approaches based on microbiome composition and genetic predisposition may enhance the effectiveness of Mediterranean diet-based strategies for managing allergic disease.

#### 5.2.3. Breastfeeding and Allergies: The Role of Early Nutrition in Immune Development

Breastfeeding is universally acknowledged as the best method of infant feeding, supplying essential nutrients, growth factors, and immune-modulating components crucial for an infant’s developing immune system [166]. Unlike other primates, human infants are born neurologically and physiologically immature, placing significant importance on breast milk for postnatal immune programming. Breast milk is rich in immune-active compounds such as IgA antibodies, antimicrobial peptides, cytokines, and soluble receptors, which collectively safeguard against infections, obesity, necrotizing enterocolitis, diabetes, and allergic diseases [167].

Despite its numerous health benefits, the efficacy of breastfeeding in preventing allergies is still subject to debate. Some studies indicate that breastfeeding can protect against asthma and allergic diseases, while others find no significant protective correlation, especially when considering genetic factors [168,169]. Epidemiological evidence suggests a reduced risk of asthma in young children linked to breastfeeding [170], but findings across different risk groups and age stages remain inconsistent, with some research failing to demonstrate a clear protective effect [171]. This suggests that breastfeeding alone might not fully prevent allergic diseases, and that factors such as maternal diet, environmental exposures, and genetic predispositions might also significantly influence outcomes.

Moreover, breastfeeding plays a critical role in shaping the infant’s gut microbiome, which is fundamental in developing immune tolerance and preventing allergies. Breastfed infants often have an early dominance of *Bifidobacterium*, a commensal bacterium linked to immune modulation and reduced allergic sensitization [172]. Additionally, the combination of breastfeeding and vaginal birth positively impacts the infant’s gut microbiota composition, enhancing microbial diversity and promoting immune resilience [173].

Breast milk also includes cytokines, inflammatory mediators, and signaling molecules that regulate immune development and may reduce the risk of allergic diseases [174]. However, the prophylactic effects of breastfeeding on allergy prevention are not definitively proven, with studies showing mixed results.

The maternal diet during lactation could affect the composition of immune factors in breast milk, thereby influencing infant immune programming. Strategies such as avoiding cow’s milk formula supplementation in the first week of life have been suggested to decrease allergy risk, although conclusive evidence is lacking [174].

While breastfeeding provides critical immune components and supports gut microbiome development, its role in allergy prevention remains uncertain due to the variability in research outcomes. Future studies should aim to identify specific maternal and infant factors that may enhance breastfeeding’s protective effects against allergies. This research could pave the way for personalized nutrition and microbiome-targeted strategies to optimize allergy prevention in early life.

#### 5.2.4. Personalized Nutrition in Food Allergy: From Diagnostics to Dietary Management

The management of food allergies (FAs) has undergone significant transformation in recent years, moving beyond generic avoidance strategies toward more individualized and evidence-based approaches. With the advent of precision medicine, a personalized nutrition framework has become increasingly important in tailoring allergen avoidance and dietary management to individual food allergy phenotypes [175,176].

Traditionally, the cornerstone of food allergy management was strict avoidance of the offending allergen, supported by emergency medication for accidental exposure. However, this “one-size-fits-all” approach is now giving way to personalized dietary strategies that consider the severity of allergic reactions, cross-reactivity, and evolving tolerance patterns. Component-resolved diagnostics (CRD), basophil activation tests, and oral food challenges (OFCs) are providing greater diagnostic precision [177]. CRD helps differentiate true allergies from cross-sensitization and can predict reaction severity by identifying IgE responses to specific allergenic epitopes.

Tailoring diets based on food allergy phenotypes enables a more flexible approach—one that may allow selective inclusion of baked or hydrolyzed forms of allergens like milk or egg, reducing dietary restrictions and improving quality of life [178]. Immunological subtypes, such as IgE- versus non-IgE-mediated allergies, as well as patient history and threshold levels, inform decisions on the degree of avoidance required [179].

Beyond immune reactivity, personalized nutrition must account for a range of factors, including age, dietary preferences, cultural and religious practices, physical activity, microbiome composition, and even epigenetic signatures. For instance, epigenetic modifications in FOXP3 and PGM3 genes influence Treg cell function and immune regulation in allergic responses [180,181]. These insights emphasize the need for integrating genetic and microbiome profiles into individualized dietary planning.

Importantly, regional differences in allergen prevalence—such as the higher incidence of cashew allergy in Southeast Asia compared to peanut allergy in the West—should guide allergen-specific dietary recommendations [182,183]. While the “big eight” allergens remain broadly relevant—milk, egg, fish, shellfish, peanuts, tree nuts, soy, and wheat—local patterns of consumption and sensitization must be considered [184].

Personalized nutrition in food allergy also involves education and behavior change. Patients and families must navigate food labels, prevent cross-contamination, and adapt recipes without compromising nutritional adequacy. This is especially critical in children, where elimination diets can lead to growth impairment and micronutrient deficiencies if not properly managed [176,177]. Registered dietitians play a key role in balancing allergen avoidance with nutritional sufficiency.

Ultimately, personalized food allergy management integrates advanced diagnostics, immunological profiling, patient preferences, and local dietary patterns into an individualized care plan. This precision approach not only improves dietary adherence and nutritional status but also supports tolerance development and reduces the psychosocial burden associated with food allergies.

The management of food allergies is shifting from standardized allergen avoidance to a precision nutrition approach tailored to individual phenotypes and immunological profiles. This paradigm integrates advanced diagnostic tools—such as component-resolved diagnostics and oral food challenges—with patient-specific variables, including age, epigenetic markers, microbiome composition, cultural dietary habits, and regional allergen exposure. Such personalized strategies enhance clinical safety, ensure nutritional adequacy, and promote immune tolerance, thereby mitigating the psychosocial impact of food allergies. By facilitating immune regulation and improving quality of life, precision nutrition represents a significant advancement in the field of allergy treatment.

Table 6 summarizes some practical recommendations and considerations for implementing personalized nutrition strategies in the management of food allergies.

### 5.3. Breaking the Cycle: How Nutrition Can Combat Immune Dysregulation in Multimorbidity

Multimorbidity, the simultaneous presence of two or more chronic diseases within an individual, presents a significant public health challenge, especially in aging populations [185]. Key lifestyle factors such as smoking, prolonged sedentary behavior, and obesity have been identified as significant risk factors for developing multimorbidity [186]. Among these, dietary patterns stand out as crucial determinants of multimorbidity risk, with evidence suggesting that nutritional interventions could play a pivotal role in both disease prevention and management [187].

The intricate link between multimorbidity and immune dysregulation is mediated through several key mechanisms, with chronic inflammation acting both as a consequence and a driver of various disease processes. Immune dysregulation often manifests as persistent, low-grade inflammation, leading to widespread tissue damage and contributing to the development of multiple chronic diseases concurrently [188].

This dynamic is significantly impacted by aging, which is linked to immunosenescence—a gradual weakening of the immune system, reducing vaccine effectiveness and raising infection susceptibility. Paradoxically, the aging process can also heighten the risk of autoimmunity due to the weakening of immune regulation and the impairment of tolerance mechanisms. Such immune aging processes can accelerate the progression of age-related conditions, thereby intensifying the link between multimorbidity and immune dysfunction [189].

A primary driver of immune dysregulation is the abnormal production of cytokines, with pro-inflammatory signaling adversely affecting multiple organ systems. This disruption in cytokine balance has been implicated in a range of diseases, including cardiovascular disorders, diabetes, and neurodegenerative diseases. Furthermore, the gut microbiome plays a pivotal role in overall immune function, with dysbiosis contributing significantly to immune dysfunction. When this microbial imbalance is combined with increased intestinal permeability, also known as “leaky gut”, bacterial components can enter the bloodstream, triggering systemic inflammation and fostering the onset of chronic diseases [190].

Metabolic dysregulation is another critical factor influencing immune function. In conditions such as obesity and diabetes, metabolically active tissues, such as adipose tissue, release inflammatory mediators that exacerbate systemic immune dysregulation [191]. Additionally, oxidative stress, marked by an overproduction of reactive oxygen species (ROS), causes cellular and tissue damage, further fueling immune dysregulation. This oxidative burden is a key contributor to the progression of multimorbidity, highlighting the crucial role of nutrition in modulating inflammation and reducing the overall disease burden [192].

Several studies have explored the impact of diet on multimorbidity. For example, the United Kingdom Women’s Cohort Study, which included 25,389 women aged 35–69 years and followed them over a median period of 22 years, investigated the associations between nutrient intake and the risk of multimorbidity, assessed using the Charlson Comorbidity Index derived from hospital records. The study found that higher daily intakes of energy and protein were associated with an increased risk of multimorbidity by 8% and 12%, respectively. Conversely, greater consumption of vitamin C and iron was linked to a slightly reduced risk, with iron showing particularly protective effects among women under 60 years of age. Interestingly, higher intakes of vitamin B12 and vitamin D were initially associated with increased risk; however, these associations lost statistical significance in sensitivity analyses. Overall, the findings underscore the complex and individualized effects of nutrient intake on multimorbidity risk and highlight the importance of developing personalized dietary guidelines [193]. A longitudinal study involving 1020 Chinese participants examined the impact of nutrition on the progression of multimorbidity over a five-year period. During this time, the prevalence of multimorbidity increased from 14% to 34%. Higher consumption of fruits, vegetables, and grain products other than rice and wheat was associated with healthier trajectories in multimorbidity progression. These foods were linked to greater intakes of dietary fiber, iron, magnesium, and phosphorus, which were also correlated with improved health outcomes. This study provides supporting evidence that a diet rich in fruits, vegetables, and whole grains may help mitigate the risk and progression of multimorbidity [194]. A cross-sectional study [195] of 129,369 Dutch adults from the Lifelines cohort investigated the associations between dietary patterns and multimorbidity within cardiometabolic domains. Four distinct dietary patterns were identified: (1) meat, alcohol, and potato; (2) snack; (3) bread and sweets; and (4) vegetable, fish, and fruit. The findings revealed that higher adherence to the meat, alcohol, and potato pattern, as well as the snack pattern, was associated with an increased prevalence of multimorbidity, particularly among men. In contrast, adherence to the bread and sweets pattern and the vegetable, fish, and fruit pattern appeared to have a protective effect; however, the latter association diminished after adjusting for body mass index (BMI). These results underscore the potential of targeted dietary interventions to prevent and manage multimorbidity through the promotion of healthier eating patterns. A prospective cohort study of 348,290 participants from the UK Biobank investigated the associations among dietary patterns, specific food group consumption, and the risk of multimorbidity over a median follow-up period of eight years [196]. Three distinct dietary patterns were identified: Western, White Meat, and Prudent. The Western pattern was associated with an increased risk of multimorbidity (HR Q5 vs. Q1 = 1.06, 95% CI: 1.03–1.09), while moderate adherence to the White Meat pattern (HR Q3 vs. Q1 = 0.97, 95% CI: 0.94–0.99) and high adherence to the Prudent pattern (HR Q5 vs. Q1 = 0.92, 95% CI: 0.90–0.95) were linked to reduced risk. Additionally, frequent consumption of processed meat and poultry was associated with a higher risk of multimorbidity, whereas greater intake of fish, fruits, and cereals showed a protective effect. These findings underscore the critical role of dietary patterns in the prevention and management of multimorbidity. Burton et al. [197], through the InCluSilver project, highlighted nutrigenomics as a pivotal tool in deciphering complex diet–genotype interactions. Their research illustrates how understanding genetic predispositions can inform dietary interventions, particularly in aging populations dealing with metabolic and degenerative diseases. They advocate for integrating advanced nutritional monitoring technologies—such as real-time tracking via wearable devices—to enhance adherence and clinical effectiveness.

The MD has garnered attention for its potential protective effects against multimorbidity. For example, a cross-sectional study of 1140 Cypriot adults explored the relationship between adherence to the Mediterranean diet and the presence of multimorbidity [198]. The results showed that higher adherence to the Mediterranean diet was significantly associated with lower odds of multimorbidity, even after adjusting for age, gender, smoking status, and physical activity (OR = 0.68, 95% CI: 0.46–0.99). Notably, men and rural residents demonstrated greater adherence compared to women and urban residents. These findings underscore the potential protective role of the Mediterranean diet against multimorbidity and support its inclusion in national dietary guidelines and public health strategies. Another study involving 143 geriatric patients (mean age: 73.1 years) examined the relationship among adherence to the Mediterranean diet, multimorbidity, and depressive symptoms [199]. The findings revealed that higher adherence to the MD was significantly associated with lower scores for both depressive symptoms and multimorbidity, even after adjusting for potential confounders. Mediation analysis further indicated that the Mediterranean diet partially mitigated the impact of multimorbidity on depressive symptoms. These results highlight the potential role of the MD in supporting mental well-being and promoting healthy aging among older adults.

Together the above suggest that the MD is associated with a lower risk of developing multiple chronic conditions and may also offer protective effects against depressive symptoms in older adults with multimorbidity. Additionally, a dietary pattern rich in white meat, particularly fish and poultry, appears to be linked to reduced multimorbidity risk; however, frequent consumption of poultry alone may increase disease risk, highlighting the nuanced and complex nature of dietary influences on health.

Despite these promising findings, most existing evidence remains cross-sectional, and comprehensive longitudinal cohort studies, especially those exploring the link between specific nutrient intakes and multimorbidity, are limited. While dietary influences on individual chronic conditions have been widely studied, fewer investigations have addressed diet’s role in multimorbidity as an interconnected whole. Poor nutrition can serve both as a driver and consequence of multimorbidity.

Nutrient deficiencies—often stemming from diets rich in processed and nutrient-poor foods—play a direct role in the development of chronic diseases, including ischemic heart disease, cerebrovascular disease, various cancers, diabetes, and Alzheimer’s disease [200]. Diets dominated by processed foods not only elevate systemic inflammation but also aggravate immune dysregulation. This occurs through deficiencies in essential micronutrients such as iron, vitamin B12, folate, and calcium.

Individuals with multimorbidity frequently experience poor nutritional status. This is commonly due to factors such as the burden of chronic disease, reduced appetite, medication side effects, and gastrointestinal dysfunction. These factors form a vicious cycle in which chronic conditions impair nutrient absorption and utilization, thereby worsening immune function and overall health [201].

A growing body of evidence emphasizes the critical role of nutrition in modulating the molecular mechanisms underlying immune aging and multimorbidity [202]. Beyond supplying essential vitamins and minerals, nutrients influence cellular processes such as mitochondrial efficiency, oxidative stress balance, and epigenetic regulation. For instance, magnesium, B-vitamins, and coenzyme Q10 improve mitochondrial function and reduce reactive oxygen species (ROS), thereby mitigating inflammation associated with aging. Similarly, folate, vitamin B12, choline, and dietary polyphenols serve as epigenetic regulators that influence DNA methylation and may slow biological aging. In addition, anti-inflammatory nutrients—such as omega-3 fatty acids, resveratrol, curcumin, and SCFAs derived from dietary fiber—can suppress inflammasome activation and cytokine overproduction, thereby promoting immune tolerance and metabolic balance. Micronutrients like zinc, selenium, vitamin D, and vitamin C are also essential for maintaining T cell function, immune surveillance, and protection against chronic inflammation [203].

Together, these findings underscore the importance of strategic dietary interventions. Specifically, nutrient-dense, anti-inflammatory diets—such as the Mediterranean diet (MD)—may help attenuate immunosenescence, reduce the risk of multimorbidity, and support healthy aging.

As scientific insights into the immunological mechanisms of multimorbidity and allergic diseases deepen, dietary and microbial interventions emerge as complementary strategies to traditional treatments. Targeted dietary patterns and nutrient supplementation can directly modulate immune cell activity, reduce chronic inflammation, and restore gut microbiota balance. Approaches such as high-fiber consumption, Mediterranean dietary practices, and targeted micronutrient supplementation effectively address both the root causes (e.g., gut dysbiosis, oxidative stress) and the downstream consequences (e.g., cytokine imbalance, immune senescence) of chronic disease clustering. Integrating these nutritional strategies into broader preventive and therapeutic frameworks represents a promising pathway to reduce the incidence and severity of immune-mediated and inflammatory diseases throughout the lifespan [204].

#### Obesity, Aging, and Multimorbidity: The Role of Precision Nutrition

As previously discussed, multimorbidity represents a significant public health challenge, affecting up to 95% of the primary care population aged 65 years and older. Although its prevalence is rising, the risk factors driving multimorbidity remain incompletely understood.

Aging is widely recognized as the primary risk factor, with recent studies suggesting that it reflects a progressive decline in physiological reserve and function, often accompanied by chronic low-grade inflammation, hormonal dysregulation, and increased vulnerability to chronic disease [65]. However, aging alone does not fully explain the condition’s complex etiology. Demographic and socioeconomic factors also contribute significantly, with women and individuals of lower socioeconomic status being particularly at risk, though the mechanisms underlying these disparities are still unclear (Reviewed in [205]).

Obesity has emerged as a major contributor to multimorbidity [206,207,208]. The accumulation of multiple unhealthy lifestyle factors further elevates this risk [194], especially in middle-aged adults who experience both obesity and multimorbidity at higher rates than older adults [209]. Given their longer life expectancy, this group faces a prolonged burden of chronic illness and related complications. Obesity is linked to the development of numerous chronic diseases [210], and imposes substantial healthcare costs [211]. Moreover, multimorbidity patterns involving obesity are frequently associated with social isolation, increased vulnerability [212], poorer health outcomes, and higher hospitalization rates [213].

Addressing obesity-related multimorbidity requires an integrated strategy that includes lifestyle changes, targeted nutrition, early detection, and public health initiatives. Dietary interventions focused on weight reduction and metabolic health—particularly those emphasizing fruits, vegetables, whole grains, and lean proteins—can reduce inflammation and counteract obesity-related metabolic disruptions [214]. Nutrients with anti-inflammatory properties, such as omega-3 fatty acids, fiber, and antioxidants, also help modulate immune responses and mitigate chronic inflammation [215]. When combined with physical activity and smoking cessation, these strategies enhance overall health and reduce multimorbidity risk. Public health campaigns promoting healthy diets and active lifestyles further support prevention and long-term behavior change. Collectively, these measures aid not only in the management of existing conditions but also in the prevention of new diseases, ultimately helping to break the cycle of multimorbidity and improve public health outcomes.

With the global rise in multimorbidity, conventional dietary recommendations often fall short of addressing the complex needs of individuals managing multiple chronic conditions simultaneously. Personalized nutrition (PN), guided by the principles of precision medicine, offers a more effective alternative. By integrating individual-specific factors—such as genetic and epigenetic profiles, microbiome composition, and lifestyle characteristics—PN enables the development of customized dietary strategies tailored to each person’s unique health context. Emerging evidence supports the potential of these personalized dietary interventions to reduce the risk and burden of multimorbidity and improve associated health outcomes [216].

Implementing PN in the context of multimorbidity requires more than general dietary guidance; it demands the integration of multiple individual-specific variables to address the unique challenges posed by coexisting chronic diseases. Factors such as genetic predisposition, age, sex, dietary habits, cultural and socioeconomic background, microbiome composition, and disease severity all shape an individual’s nutritional needs and therapeutic response. For example, patients with heart failure must manage sodium and fluid intake to prevent decompensation, while individuals with neurodegenerative diseases often face dysphagia and altered nutrient absorption, requiring tailored dietary strategies [217]. These examples highlight the inadequacy of conventional dietary recommendations in addressing the complexity of multimorbidity.

Recent advances in genetics and epigenetics have strengthened the scientific foundation for PN. Genetic polymorphisms influence how nutrients are metabolized and utilized, directly affecting disease susceptibility and progression. At the same time, epigenetic mechanisms—such as DNA methylation and histone modifications—respond dynamically to dietary inputs, modulating gene expression in ways that influence long-term health. By incorporating these molecular insights, PN enables clinicians to formulate more precise and effective nutritional interventions [218].

The gut microbiome represents another critical dimension in PN. As a central regulator of inflammation, immunity, and metabolism, the microbiome plays a pivotal role in the development and management of chronic conditions commonly seen in multimorbid patients—including obesity, type 2 diabetes, cardiovascular disease, and neurodegenerative disorders [219]. Personalized interventions aimed at restoring microbial balance—such as prebiotic and probiotic supplementation, fermented foods, and high-fiber diets—can improve metabolic function and reduce systemic inflammation, offering tangible benefits for patients with multiple coexisting conditions [220].

Moreover, recent technological advancements provide the practical tools needed to apply PN in real-world settings. AI-powered platforms, wearable sensors, and mobile health applications facilitate real-time dietary assessment, personalized feedback, and adaptive nutrition planning. These innovations empower healthcare providers and patients to co-manage dietary strategies dynamically, improving adherence, engagement, and outcomes [221].

Together, these emerging tools and scientific advances establish PN not just as a theoretical model, but as a practical, necessary solution to the challenges of multimorbidity. By integrating molecular data, gut health, and real-time monitoring, personalized nutrition represents a paradigm shift toward more precise, effective, and sustainable chronic disease management.

However, the broad implementation of personalized nutrition faces practical challenges, including accessibility, cost-effectiveness of advanced diagnostics and nutritional monitoring tools, and the need for specialized training for healthcare providers [222]. Ethical issues related to data privacy, patient consent, and equitable access to technologies and resources must also be thoughtfully addressed. Ensuring culturally sensitive, inclusive, and equitable dietary approaches is paramount for successful integration into routine healthcare [223]. To overcome these challenges, practical recommendations include the following:Conducting comprehensive individualized nutritional assessments incorporating genetic, microbiome, and advanced dietary monitoring data.Integrating AI-driven tools, wearables, and mobile health applications into standard clinical practice.Fostering interdisciplinary collaboration among nutritionists, geneticists, microbiologists, healthcare professionals, and technologists.Prioritizing patient-centered care by respecting individual preferences, cultural factors, lifestyle considerations, and accessibility concerns in personalized dietary interventions.

Future research should prioritize rigorous longitudinal studies and clinical trials that validate precision nutrition’s clinical benefits across diverse populations. Continued development in multi-omics analyses, bioinformatics, and machine learning algorithms will further refine personalized dietary strategies. Additionally, comprehensive training programs for healthcare providers will be essential to integrate these sophisticated approaches effectively into clinical routines. Personalized nutrition, thus, represents a transformative advancement in multimorbidity management, promising targeted, effective interventions attuned to individual biological, environmental, and behavioral contexts. Embracing these precision dietary strategies will significantly enhance patient outcomes, reduce the overall burden of chronic diseases, and support healthier aging across populations.

## 6. Technological Advances in Nutritional Assessment and Monitoring

Accurate nutritional assessment is foundational for effective dietary management, especially within personalized healthcare frameworks. As mentioned above, recent technological advancements—including AI, wearable sensors, mobile applications, and machine learning—are transforming nutritional assessment, offering unprecedented precision, ease of use, and real-time capabilities. These innovations facilitate tailored dietary interventions, addressing limitations of traditional methods and promoting individualized health strategies.

Conventional dietary assessment methods, such as dietary recall, food diaries, and frequency questionnaires, have inherent limitations related to accuracy, recall bias, participant burden, and scalability. AI and machine learning (ML) address these limitations by improving data collection accuracy, streamlining analysis, and enabling personalized feedback. AI-powered dietary assessment tools utilize automated image recognition, comprehensive food databases, and algorithms that estimate nutrient content with high precision [224].

Das et al. [225] highlighted the transformative potential of AI-based dietary assessment tools, emphasizing their enhanced accuracy and feasibility. Technologies like computer vision and deep learning algorithms can analyze photographic inputs of meals, providing immediate feedback on calorie intake and nutrient composition. Gelli et al. [226] further validated the utility of these technologies in real-world settings, showing high agreement between AI-assisted dietary assessments and traditional weighed records.

Wearable devices have gained widespread acceptance in physical activity monitoring and are now increasingly utilized for dietary assessment. Motion sensors, accelerometers, gyroscopes, and smartwatches can passively track eating behaviors, identifying food consumption through wrist movements, chewing patterns, and physiological markers. A systematic review by Heydarian et al. [227] confirmed the promising accuracy of wrist-worn sensors for monitoring eating activities, particularly when integrated with machine learning models such as Support Vector Machines (SVMs), Random Forest, Hidden Markov Models (HMMs), and deep learning algorithms.

These wearable tools offer a non-intrusive, real-time dietary monitoring approach, enhancing adherence to nutritional interventions and supporting behavior modification programs. Additionally, they enable data collection in free-living conditions, reflecting true dietary habits more accurately than traditional methods.

Mobile applications have revolutionized nutritional assessment by leveraging visual recognition technology and user-friendly interfaces. These applications employ advanced algorithms to recognize food items from smartphone-captured images, automatically estimate portion sizes, and calculate nutrient content. Such tools significantly reduce recall bias and improve dietary tracking precision.

Recent studies [226,228,229,230] highlight the efficacy and accuracy of mobile-based dietary assessment tools, particularly in low-resource settings. The convenience, accessibility, and immediacy of mobile technology have significant implications for broadening the reach of precise dietary interventions globally.

Despite significant advances, several challenges remain in implementing technological tools for nutritional assessment. Issues of data privacy and security, particularly with sensitive health-related information, are major concerns. Ethical considerations regarding data ownership, informed consent, and equitable access to technology also demand attention. Additionally, disparities in technological literacy and accessibility could potentially exacerbate health inequities if not proactively addressed [231].

Another critical challenge involves maintaining updated and accurate food composition databases essential for precise nutritional assessment. Variability in food composition across regions and cultures necessitates regular updates and tailored interfaces to ensure cultural and nutritional relevance [232].

Practical applications of technology-enhanced nutritional assessment are expanding rapidly across clinical, research, and public health settings. For example, AI-driven mobile apps have been employed in clinical trials to assess dietary adherence in real-time, providing feedback to participants and researchers alike. Similarly, wearable devices are utilized in chronic disease management, helping patients monitor dietary habits and physical activity, thus enhancing adherence to personalized nutritional interventions.

In community settings, these technological tools facilitate public health initiatives, supporting large-scale dietary monitoring and intervention programs aimed at improving nutritional behaviors and health outcomes across diverse populations.

Looking forward, continued innovation in technology-enhanced nutritional assessment is essential. Key priorities include the following:Enhancing AI Accuracy and Usability: Refine AI algorithms, improve machine learning models, and enhance user interfaces to optimize accuracy, reliability, and user experience.Interdisciplinary Collaboration: Foster collaboration among nutritionists, clinicians, software developers, data scientists, and policymakers to design integrative and user-centered nutritional monitoring platforms.Reducing Health Disparities: Develop affordable and accessible technologies, ensuring their applicability across diverse socioeconomic groups and resource-limited settings.Ensuring Ethical and Transparent Data Management: Strengthen regulatory frameworks and ethical guidelines to protect user privacy, data security, and informed consent.Validation Studies and Real-world Testing: Conduct robust validation studies across various populations and settings, including longitudinal assessments, to verify reliability, effectiveness, and clinical utility of these technologies.

Together, the above suggest that technological advancements represent a transformative force in nutritional science, enabling precision nutrition approaches that were previously unattainable. These tools significantly enhance the ability of healthcare professionals to accurately assess, monitor, and guide nutritional interventions tailored to individual needs. By overcoming traditional assessment limitations, technology-driven nutritional monitoring offers unparalleled potential to prevent and manage chronic diseases, optimize immune health, and ultimately improve patient outcomes across diverse populations.

## 7. Conclusions, Recommendations, and Perspectives

The escalating global prevalence of allergic diseases and multimorbidity underscores the imperative for innovative approaches in healthcare. This comprehensive review highlights the intricate and bidirectional relationships among nutrition, gut microbiota, immune regulation, allergic responses, and chronic disease management. Diet has emerged not simply as a passive factor in health, but as an active modulator of immune responses, influencing disease susceptibility and progression. Nutritional strategies, including personalized dietary interventions, early-life microbiome modulation, and adherence to anti-inflammatory dietary patterns such as the MD, offer substantial promise for enhancing immune resilience, reducing systemic inflammation, and ultimately improving patient outcomes.

The dietary and nutritional strategies discussed in this review are summarized in Table 7. These evidence-based interventions outline key mechanisms and associated health benefits, highlighting practical approaches for clinicians and public health professionals to improve immune resilience, reduce inflammation, and manage multimorbidity and allergic disease.

Advances in diagnostic precision—including CRD, basophil activation tests, oral food challenges, and epigenetic biomarker profiling—are facilitating more accurate and individualized approaches to allergy management. Technological advancements, such as AI, wearable sensors, and mobile dietary tracking applications, are further revolutionizing nutritional assessment and intervention, making personalized care feasible on a broader scale. Based on the current evidence, we recommend the following strategies for clinicians, researchers, and policymakers:Integration of personalized nutrition into clinical care: Adopt personalized dietary recommendations informed by patient phenotype, microbiome profiles, genetic background, lifestyle, cultural preferences, and local dietary patterns. Such precision nutrition plans should aim to balance nutrient adequacy, allergen avoidance, and quality of life.Leverage technological innovations: Incorporate AI-based tools, wearable sensors, mobile applications, and real-time dietary monitoring technologies into clinical practice and public health programs. This integration can enhance accuracy in dietary assessment, adherence to nutritional guidance, and support continuous patient engagement.Promotion of anti-inflammatory dietary patterns: Prioritize dietary patterns such as the Mediterranean diet for their proven capacity to enhance immune resilience, modulate microbiota composition, and reduce chronic inflammation. Public health policies should support widespread education and accessibility of such dietary models.Life-course approach to nutrition and immune health: Nutritional interventions should begin prenatally, continue through childhood, adulthood, and into older age, to build long-term immune resilience and delay the onset or progression of chronic diseases.Ensuring equitable access to personalized care: Develop inclusive and culturally sensitive nutrition programs and digital health solutions to ensure that personalized nutritional strategies are accessible and affordable across diverse socioeconomic and geographic populations, preventing further disparities in healthcare.

Despite substantial advances, several critical gaps and challenges persist:Long-term data and interventional studies: There is a clear need for robust, long-term randomized controlled trials and cohort studies to assess the sustained impacts of personalized nutrition and microbiome-based interventions on multimorbidity and allergic disease progression.Standardization of methodologies and data integration: Heterogeneity in dietary assessment methods, microbiome analyses, and nutritional biomarker interpretation remains a challenge. Efforts to standardize methodologies and integrate datasets will enhance comparability and clinical translation.Ethical, privacy, and accessibility issues: Data privacy, security, and ethical considerations related to digital health tools must be addressed. Additionally, ensuring equitable access to advanced nutritional technologies and personalized care remains a major public health challenge.

The future of nutrition science and immune health management is firmly rooted in precision and personalization, enabled by technological advancements and deeper biological insights. Several promising avenues should be pursued:Integration of multi-omics approaches: Combining genomics, transcriptomics, proteomics, metabolomics, and microbiome data will enhance the precision of dietary interventions and improve our understanding of individual responses to nutritional changes.Enhanced role of digital health in nutritional management: Future innovations in AI, deep learning, and wearable technologies are expected to refine dietary assessment, providing personalized feedback in real-time and improving dietary adherence and outcomes.Microbiome-based therapeutics and precision probiotics: The advancement of precision probiotics and microbiome-targeted therapies, informed by individual microbiota profiles and immune status, presents an exciting frontier for personalized nutritional interventions.Policy and public health innovations: Public health strategies should evolve to support personalized nutrition initiatives. This includes investing in community education programs, supporting food system reforms, and developing regulatory frameworks for digital health interventions.

Ultimately, achieving these future directions will require collaborative efforts among nutritionists, immunologists, clinicians, data scientists, policymakers, and community stakeholders. Such interdisciplinary collaboration will ensure that the benefits of precision nutrition and microbiome research reach all individuals, enhancing immune health, preventing chronic diseases, and improving quality of life across diverse populations.

Future studies must adopt integrative, system-level approaches to fully capture the complexity of host–microbiome interactions. In particular, the integration of data from gut microbiota metabolomics, host transcriptomics, epigenomics, and immune profiling offers a powerful framework for elucidating the molecular networks that regulate immune homeostasis and disease progression. These multi-omics strategies enable researchers to trace the cascade of biological events—from nutrient metabolism to inflammatory responses—at unprecedented resolution, uncovering how diverse molecular pathways converge to influence immune function and clinical outcomes.

Applying such models is especially critical for disentangling the intricate relationships underlying conditions like allergies and multimorbidity. By integrating high-dimensional datasets, investigators can identify novel biomarkers, mechanistic pathways, and therapeutic targets. For example, subtle shifts in microbiota-derived metabolites may correlate with specific host transcriptional signatures that modulate immune cell activation and inflammatory pathways. Elucidating these connections will not only refine our understanding of disease etiology but also facilitate the development of predictive models for patient response to nutritional and pharmacological interventions.

There is also a growing need for cross-cultural research that extends beyond conventional laboratory settings. Investigating nutritional interventions across diverse populations and dietary traditions enhances the generalizability of findings and uncovers context-dependent interactions. Cultural and dietary heterogeneity serves as a natural experiment, offering insight into how different nutritional environments shape immune and metabolic responses. When coupled with multi-omics analyses, such research can inform the development of culturally appropriate, personalized dietary strategies tailored to individual genetic and environmental profiles.

Moreover, the inclusion of longitudinal and interventional study designs will be vital for establishing causal relationships and understanding how dietary interventions influence the host–microbe axis over time. Multidisciplinary collaboration—bringing together clinicians, microbiologists, nutrition scientists, and computational biologists—will be essential for translating molecular findings into actionable, patient-centered interventions. Advances in computational modeling and machine learning are already playing a pivotal role in integrating complex datasets, setting the stage for dynamic and responsive precision nutrition strategies.

Ultimately, by combining multi-omics technologies with culturally informed research designs, the field can move toward a truly integrative understanding of allergies and multimorbidity. This paradigm shift holds the potential to transform clinical practice, enabling personalized nutritional interventions that are proactive, predictive, and deeply rooted in the complexity of human biology and culture. Such an approach represents a significant step forward in preventive medicine and has the potential to improve health outcomes and quality of life on a global scale.

## 8. Limitations

This review is narrative in nature and does not adhere to systematic or scoping review frameworks. Although structured search strategies were implemented and SANRA criteria guided methodological rigor, we did not conduct quantitative synthesis or formal risk-of-bias assessments. Consequently, selection bias may have influenced study inclusion, potentially resulting in overrepresentation of published positive findings. This bias can be attributed to our reliance on databases that favor positive results, limited inclusion of gray literature, and potential omission of relevant non-indexed studies.

The heterogeneity included studies—spanning basic science, clinical trials, and epidemiological research—inherently limits comparability and generalizability. Substantial variations in study design, population characteristics, interventions, and outcome measures complicated the synthesis of results and may constrain definitive conclusions. Additionally, the rapidly evolving nature of microbiome science and precision nutrition presents challenges in achieving comprehensive coverage of emerging evidence.

Furthermore, narrative reviews necessarily sacrifice some methodological rigor characteristic of systematic reviews, including predetermined inclusion/exclusion criteria, comprehensive search strategies, and quality appraisal using standardized instruments. This introduces an element of subjectivity in evidence selection and interpretation. Despite these limitations, we have endeavored to present an extensive cross-disciplinary synthesis that both reflects current knowledge and identifies critical directions for future research. Subsequent investigations should employ systematic methodologies to enhance the robustness and reproducibility of findings in this field

## Figures and Tables

**Figure 1 nutrients-17-01685-f001:**
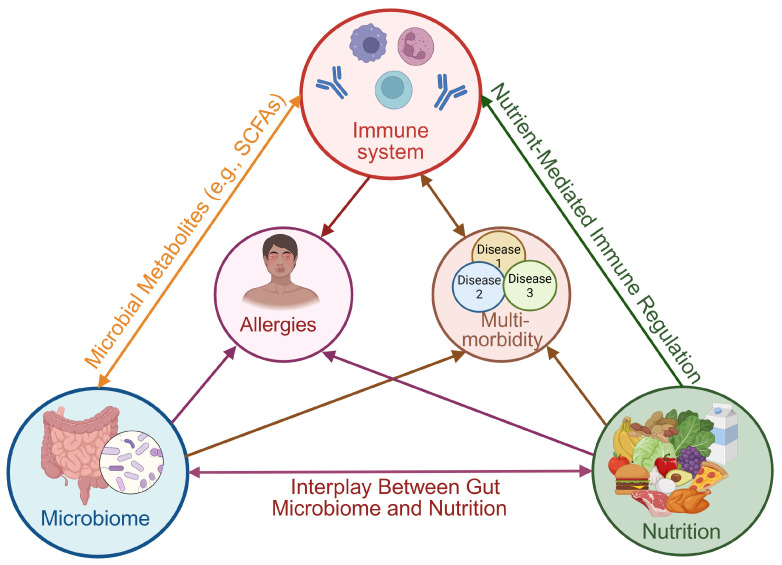
The dynamic interplay among nutrition, the gut microbiota, and the immune system—collectively termed the “triad of resilience”—and its role in modulating allergies and multimorbidity. Dietary inputs shape the gut microbiome, which in turn produces metabolites such as short-chain fatty acids (SCFAs) that regulate immune function. Disruptions in this interconnected system can lead to immune dysregulation, contributing to chronic inflammatory conditions and allergic diseases. Overall, the figure illustrates the critical relationship among nutrition, microbiota, and immune balance in supporting health and resilience. Figure created using BioRender (https://BioRender.com; accessed on 13 April 2025).

**Table 1 nutrients-17-01685-t001:** Key differences between innate and adaptive immunity ^1^.

Feature	Innate Immunity	Adaptive Immunity
Feature	Innate immunity	Adaptive immunity
Response Time	Immediate (mins to hours)	Delayed (days to weeks)
Specificity	Non-specific, recognizes conserved patterns (PAMPs)	Highly specific, targets unique antigens
Memory	No memory	Generates immunologic memory
Key Cells	Macrophages, neutrophils, NK cells, dendritic cells	B cells, T cells (CD4^+^, CD8^+^)
Soluble Factors	Complement, cytokines, and antimicrobial peptides	Antibodies, cytokines

^1^ Data obtained from [32,33].

**Table 2 nutrients-17-01685-t002:** Key immunological players in allergic responses ^1^.

Component	Role in Allergic Response
Antigen-Presenting Cells (APCs)	Capture and present allergens to naïve T cells, initiating adaptive immune responses.
T-helper 2 (Th2) Cells	Drive allergic inflammation by secreting IL-4, IL-5, and IL-13, enhancing IgE production and eosinophil activity.
Regulatory T Cells (Tregs)	Suppress excessive immune responses; impaired Treg activity contributes to allergic sensitization.
B Cells and Plasma Cells	Produce allergen-specific IgE antibodies that sensitize mast cells and basophils to allergens.
Mast Cells	Store histamine and inflammatory mediators release them upon IgE cross-linking, triggering allergic symptoms.
Basophils	Circulating cells amplify allergic inflammation through the release of histamine, leukotrienes, and IL-4.
Eosinophils	Mediate late-phase allergic responses; release cytotoxic granules, causing chronic inflammation and tissue damage.
Cytokines	IL-4: Induces IgE class switching. IL-5: Recruits eosinophils. IL-13: Promotes mucus production and airway remodeling. IL-10 and TGF-β: Anti-inflammatory cytokines, reduced levels contribute to allergy.
Histamine and Leukotrienes	Released from mast cells and basophils; mediate vasodilation, bronchoconstriction, and mucus production.
Gut Microbiota	Essential for immune tolerance; microbial dysbiosis increases susceptibility to allergic diseases.

^1^ Data obtained from [48,49].

**Table 5 nutrients-17-01685-t005:** Mediterranean diet components and their immune benefits ^1^.

Mediterranean Diet Component	Immune and Allergy-Related Effects	Dietary Sources
Olive Oil	Rich in polyphenols; anti-inflammatory; supports gut microbiota diversity and barrier integrity.	Extra virgin olive oil
Fruits and Vegetables	High in antioxidants (vitamins C, E, carotenoids); reduces oxidative stress and inflammation.	Leafy greens, berries, citrus fruits, tomatoes, bell peppers, carrots
Whole Grains	Prebiotic fiber supports gut microbiota; enhances SCFA production for immune modulation.	Whole wheat, brown rice, quinoa, barley, oats
Legumes	High fiber and protein content; contributes to gut microbiota balance and Treg activation.	Lentils, chickpeas, black beans, kidney beans
Nuts and Seeds	Source of polyphenols and healthy fats; modulates inflammatory and immune responses.	Almonds, walnuts, flaxseeds, chia seeds, sunflower seeds
Fish and Seafood	High in omega-3 fatty acids (EPA and DHA); reduces airway inflammation and asthma risk.	Salmon, sardines, mackerel, tuna, shrimp
Dairy (Moderate Consumption)	Source of probiotics may support gut microbiota and immune tolerance.	Yogurt, cheese, kefir
Red Wine (Moderate Consumption)	Contains polyphenols (resveratrol); potential immunomodulatory and anti-inflammatory effects.	Red wine (in moderation, as part of a balanced diet)

^1^ Data obtained from [127,153].

**Table 6 nutrients-17-01685-t006:** Practical recommendations for personalized nutrition in food allergy management.

Recommendation	Description
Integrate clinical phenotype and diagnostic tools	Use skin prick tests (SPTs), specific IgE levels, component-resolved diagnostics (CRD), and oral food challenges (OFCs) to define allergy phenotype and assess severity.
Use epigenetic biomarkers where available	Incorporate emerging biomarkers such as FOXP3 and PGM3 methylation status to inform immune tolerance capacity and personalize intervention strategies.
Account for regional allergen profiles	Tailor dietary advice based on geographic allergen prevalence (e.g., cashew allergy in Southeast Asia vs. peanut allergy in Western countries), food availability, and cultural dietary norms.
Tailor allergen avoidance based on risk	Adapt the strictness of avoidance diets according to reaction threshold, allergen form (raw vs. baked), and cross-reactivity risk.
Monitor and support nutritional adequacy	Ensure diets do not lead to micronutrient deficiencies, especially in children. Monitor growth and dietary intake regularly.
Incorporate microbiome insights	Consider microbiome composition and diversity (e.g., SCFA production, Bifidobacterium abundance) when planning interventions that support immune tolerance.
Apply shared decision-making and cultural sensitivity	Engage patients in personalized planning, accounting for food preferences, religious or cultural practices, and psychosocial factors to improve adherence.
Involve registered dietitians in care plans	Dietitians should provide individualized guidance on allergen avoidance, nutrient adequacy, recipe substitutions, and label reading.

**Table 7 nutrients-17-01685-t007:** Dietary and nutritional strategies for enhancing immune resilience and managing allergies and multimorbidity.

Strategy	Mechanism	Potential Benefits
High-Fiber Diet (↑ ^1^ SCFAs)	Modulates Th1/Th2 balance, enhances Tregs	↓ ^2^ Allergy risk, ↓ Inflammation
Mediterranean Diet	Rich in polyphenols, omega-3s, and fiber	↓ Asthma, eczema, CVD, depression
Breastfeeding	Promotes beneficial microbes (e.g., Bifidobacterium), provides IgA	↓ Early-life allergy susceptibility
Probiotics and Prebiotics	Restores microbial diversity, enhances gut barrier	Adjunct therapy in allergy and metabolic diseases
Targeted Micronutrient Supplementation	Corrects deficiencies that impair immune signaling	Immune rejuvenation in aging, ↓ Multimorbidity
Anti-inflammatory Compounds (e.g., turmeric, flavonoids)	Suppress NF-κB and other inflammatory pathways	Supports immune tolerance, prevents escalation of chronic disease
Intermittent Fasting/Caloric Restriction	Enhances immune cell renewal, ↓ oxidative stress	May slow immune aging and disease accumulation

^1^ ↑: Increase; ^2^ ↓: Decrease.

## Data Availability

This study generated no new data.

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
