# Peer review of "Boosting Immunity Through Nutrition and Gut Health: A Narrative Review on Managing Allergies and Multimorbidity"

_nutrients, 2025, doi:10.3390/nu17101685_

Round 1
Reviewer 1 Report
Comments and Suggestions for Authors
Thank you for the opportunity to review the manuscript. Overall, this is a scientifically rich and well-organized manuscript, but a few areas could benefit from stylistic refinements, structural strengthening, and improvements in clarity.
General Comments
Some long sentences obscure meaning and could be divided or restructured.
Occasionally, the tone becomes promotional rather than critical-scientific.
Lines 927–952 please restructure, for better readability.
Lines 965–1006 please condense the sentence.
Lines 1007–1013 are unclear, please reformulate.
Lines 1014–1044 Section needs very good argumentation.
Reviewer 2 Report
Comments and Suggestions for Authors
The paper “Boosting Immunity Through Nutrition and Gut Health: A Narrative Review on Managing Allergies and Multimorbidity” is a review paper.
In the publication, the authors have undertaken to discuss the issue of the role of the gut-immune-metabolic axis in the pathogenesis and treatment of diseases.
In my opinion, the authors have presented the discussed issues in an extremely precise and detailed manner.
The publication contains 1 figure and 5 tables.
In the conclusion, the authors have provided relevant conclusions with recommendations.
In their work, the authors have included an impressive number, 210 items of current scientific reference on the discussed topic.
In my opinion, the authors fully studied the issue that is the subject of the publication.
In summary, I highly rate the work and believe that it can be qualified for publication without revisions. There is no doubt that this is one of the best review papers on the above topics.
Author Response
We sincerely thank the Reviewer for the generous and encouraging evaluation of our manuscript. We are especially grateful for the recognition of the manuscript's clarity, scientific rigor, and comprehensive coverage of the gut–immune–metabolic axis in the context of allergies and multimorbidity. Your thoughtful comments affirm the intent and depth of our work, and we appreciate your recommendation for publication without further revisions.
Reviewer 3 Report
Comments and Suggestions for Authors
The topic is timely and relevant to clinicians and researchers in immunology, nutrition, and public health, but several major concerns have arisen.
L 87-89: “Emerging evidence points to gut microbiome composition and dietary patterns as major determinants of immune balance and disease progression in individuals with multimorbidity”
Is a strong, sweeping claim and should be supported by specific citations. Please add one or two key references—ideally recent clinical or mechanistic studies.
Pages 98–100 and 125–130 each mention elements of the study’s objectives, but these fragments should be unified into a single, cohesive paragraph that clearly and concisely states the study’s purpose.
Although this work is described as a narrative review, the “supposed” Methods section (pp. 131–141) outlines a broad database search without specifying study selection criteria, quality‐assessment procedures, or strategies for addressing conflicting evidence. Regardless of review type,, please include explicit inclusion/exclusion criteria, a summary (e.g., PRISMA‐style flow chart), and, if possible, a brief assessment of study quality (risk of bias), even if narrative. This will strengthen the review’s transparency and credibility.
Section 2.1 presents well-established concepts that have long been documented in the literature and adds little new insight. I recommend replacing the lengthy narrative with a concise schematic or figure to summarize these foundational principles.
The passages in lines 68–70 and 233–235 present the same content using different citations. To eliminate redundancy and improve clarity, please consolidate these sections into a single statement and cite all pertinent references together.
In lines 317–319 the authors are actually describing the microbiota—the community of microorganisms—whereas ‘microbiome’ more broadly encompasses these organisms’ genetic material and the physicochemical environment. Consider revising to: The human microbiota consists of diverse bacteria, fungi, viruses, and archaea that reside on and within our bodies, dynamically interacting with their host and surroundings. The term “microbiome” further includes these microorganisms’ collective genomes and the physicochemical properties of their niche.
Lines 330-333: Central to these interactions is the gut microbiota, a crucial determinant of host health that influences immune function both directly, through immune cell interactions, and indirectly, through the production of microbial metabolites derived from dietary components [71], and lines 350-353: The gut microbiota represents a highly complex and dynamic microbial ecosystem with profound implications for immune regulation. Notably, gut microbiota plays a critical role in shaping mucosal immunity, antigen presentation, and the production of metabolites that influence systemic immune balance [78]. The manuscript contains considerable repetition of the same concepts; a thorough revision is needed to streamline the text and eliminate redundancy.
Section 3.1, “Allergies and Gut Health: The Microbiome–Immune Connection”, does not present any evidence demonstrating how the intestinal flora influences allergic conditions via immune mechanisms, nor does it include data on immune modulation.
In Table 3, 4 & of course 5, each series should be accompanied by its own references.
Section 4.2.1 is overly text-heavy and would be more effective if reformatted as a table or figure.
Overall, the manuscript is overly text-heavy and can be tedious to read. To improve readability and maintain narrative flow, consider converting certain sections into tables or figures to provide visual structure and guide the reader.
A key limitation is that, despite the title and introduction emphasizing gut health, microflora, and the gut microbiome, the manuscript does not include a dedicated section on how beneficial versus pathogenic gut microorganisms interact with the immune system to influence allergies and multimorbidity.
The subsection on personalized nutrition in food allergy currently concludes abruptly without a summary. It also fails to incorporate critical elements—such as epigenetic biomarkers and regional allergen profiles—into its dietary guidance. Please expand this section to include practical, phenotype-specific dietary algorithms, illustrative case studies or decision trees, and a concise set of take-home recommendations.
Please include a dedicated ‘Limitations’ section to clearly outline the study’s constraints and potential biases.
Reviewer 4 Report
Comments and Suggestions for Authors
This paper noted Boosting immunity through nutrition and gut health: A narrative review on managing allergies and multimorbidity. This review provides a comprehensive review of the relevant mechanisms, intervention strategies, and technical applications, characterized by clear logic and a well-structured format. It particularly highlights the potential of precision nutrition and the Mediterranean diet, offering valuable insights for both clinical practice and scientific research. However, as a narrative review, it exhibits insufficient methodological rigor; some conclusions lack robust support from high-quality evidence, necessitating further supplementation and refinement. The detailed comments are listed as following:
- As a narrative review, the absence of clearly defined literature selection criteria (such as time frame and exclusion criteria) may introduce selection bias. It is recommended to include a PRISMA flow diagram or provide an explanation of the selection process.
- The clinical evidence for certain intervention strategies, such as intermittent fasting and probiotics, is insufficient. It is necessary to reference additional randomized controlled trials (RCTs) or long-term cohort studies to enhance their persuasive power.
- The application of AI and wearable devices in nutritional assessment lacks specific cases (such as algorithm types, validation data). It is suggested to supplement actual application scenarios or analysis of technical limitations.
- Some paragraphs are repetitive (for instance, the connection between immunosenescence and inflammation is repeatedly mentioned in multiple chapters). It is suggested to streamline the content to avoid redundancy.
- The format of the references should be uniform (e.g., if the abbreviations of some journal names are inconsistent).
- Supplement the latest research from 2023 to 2025 (such as the recent progress in gut microbiota and neuroimmunity).
- Add "future research directions", such as gut microbiota metabolomics, multi-omics integration analysis, and cross-cultural nutritional intervention comparison, etc.
Round 2
Reviewer 4 Report
Comments and Suggestions for Authors
None, the authors respond the comments and revised it well.